# Improving access to treatment for alcohol dependence in primary care: A qualitative investigation of factors that facilitate and impede treatment access and completion

Catharine Montgomery[1]*, Pooja Saini[1], Christine Schoetensack[1], Molly McCarthy[1], Claire Hanlon[1], Lynn Owens[2,3], Cecil Kullu[4], Nadja van Ginneken[5], Melissa Rice[6], Ryan Young[5]

1 Faculty of Health, School of Psychology, Liverpool John Moores University, Liverpool, United Kingdom, 2 Institute of Systems, Molecular and Integrative Biology, University of Liverpool, Biosciences Building, Liverpool, United Kingdom, 3 Liverpool University Hospitals NHS Foundation Trust, The Royal Liverpool and Broadgreen University Hospitals, Liverpool, United Kingdom, 4 Mersey Care NHS Foundation Trust, Royal Liverpool University Hospital, Liverpool, United Kingdom, 5 Brownlow Health Central, Princes Park Health Centre, Liverpool, United Kingdom, 6 UK Parliament, Houses of Parliament, Westminster, United Kingdom

* c.a.montgomery@ljmu.ac.uk

**Data Availability Statement:** All anonymised transcripts of the qualitative interviews are available

## Abstract

### Background

Timely intervention for people with alcohol dependence in primary care is needed. Primary care services have a key role in supporting adults with alcohol dependence and require appropriate provision of services.

### Objective

To examine the perceptions of both primary care practitioners and adults with alcohol dependence regarding service provision and to describe help seeking behaviours for adults with alcohol dependence.

### Design and setting

Qualitative study consisting of semi-structured interviews with adults with alcohol dependence, healthcare professionals and staff members of specialist alcohol services who had previous or current experience in the management, treatment, or referral of adults with alcohol dependence in Northwest England.

### Method

Interviews were conducted with ten adults with alcohol dependence and 15 staff. Data were analysed thematically, applying principles of constant comparison.

on our open access institutional data repository
operndata.ljmu.ac.uk. DOI for the qualitative
interviews is here: https://doi.org/10.24377/LJMU.
d.00000157.

**Funding:** This work was supported by an NHS
Liverpool Clinical Commissioning Group Research
Capability Fund grant RCF21-22/07 to CM, PS, LO
and CK. Additional funding for qualitative research
support and participant remuneration were
provided through the Liverpool John Moores
University Policy Support Funding awarded to CM.
The funders played no role in the design, data
collection, analysis, decision to publish or
preparation of the manuscript.

**Competing interests:** The authors have declared
that no competing interests exist.

## Results

Three themes were identified following inductive thematic analysis. The first theme, *point of access* relates to current service provision being reactive rather than preventative, the stigma associated with alcohol dependence and a person's preparedness to change. The second theme identified was *treatment process and pathways* that highlights difficulties of engagement, mental health support, direct access and person-centred support. The third theme was *follow-up care* and discusses the opportunities and threats of transitional support or aftercare for alcohol dependence, signposting and peer support.

## Conclusion

There are clear opportunities to support adults with alcohol dependence in primary care and the need to increase provision for timely intervention for alcohol related issues in primary care.

## Introduction

Alcohol-related harm costs the National Health Service (NHS) £3.5 billion a year [1]. During the Covid-19 restrictions in 2020, there were 258,811 alcohol-specific hospital admissions; 6983 deaths were related to alcohol-specific causes, an increase of 20% from the previous year; alcohol liver disease deaths increased by 58% compared to baseline—the second highest number ever recorded [2]. Increasing the number of people in treatment would reduce hospital admissions, yet only 74,618 of the 586,797 people estimated alcohol dependent are in treatment [3], with alcohol having one of the largest treatment gaps worldwide [4,5].

Alcohol harms, ranging from alcohol-related disease to hospital admissions and deaths disproportionately affect those who live in areas of high deprivation, and/or those of lower socioeconomic status (SES) [6]; those in deprived areas drink less, yet suffer higher alcohol related harms (the "alcohol harm paradox" [7]). Harmful and hazardous drinking are characterised by drinking alcohol at levels that risk causing harm to physical and mental health [8]. The most harmful level of drinking–alcohol dependence–is characterised by craving, tolerance and preoccupation with drinking in the face of adverse consequences on health such as alcohol related liver disease and mental health problems [9]. Nationally, people with alcohol dependence in areas of higher social deprivation are less likely to receive the treatment they need in a timely manner through primary care providers [10]. Deprivation has also been recognised as an important factor in prevalence and treatment of alcohol dependence in previous research interrogating large databases (e.g. the Clinical Practice Research Datalink, CPRD and UK Biobank), with one study finding deprived areas have higher rates of alcohol dependence diagnoses, higher mortality, and less support for patients once diagnosed [10,11]. The primary care system in the NHS provides a first point of contact with the health care system and includes General Practitioners (GPs) of medicine, pharmacists, dentists and opticians. Over 90% of the UK population is registered with a general practitioner [12], providing general practitioners with the opportunity to assess and identify patients with alcohol use disorders [13]. However, identification in primary care is generally low [14], and only a small proportion (11.7%) of those diagnosed with alcohol dependence in primary care receive pharmacotherapy or psychosocial support within the first year, and patients in deprived areas are less likely to receive pharmacotherapy [11].

The National Institute for Health and Care Excellence (2011) guideline [9] for the diagnosis and management of alcohol use disorders in the UK recommends that all health care professionals in primary care services should be able to assess and identify harmful drinking behaviours and alcohol dependence using recognised tools, and offer brief motivational advice followed by referral for psychological interventions, pharmacotherapy and assisted outpatient or inpatient withdrawal depending on level of harmful drinking. General practitioners thus have a key role in promoting healthier lifestyles and preventing disease [15], though in these settings there are capacity (e.g. lack of time) and capability (e.g. lack of training) barriers that may impede identification of alcohol dependence [16]. Screening and interventions to address hazardous alcohol use such as the Alcohol Use Disorders Identification Test–Consumption (AUDIT-C), followed by brief advice [17] are utilised in primary care, but can be affected by infrastructure (e.g. poor IT systems), low patient awareness of their drinking behaviour, and alcohol related variables (e.g. cost, availability, normative behaviour) [18]. Existing evidence suggests that patients prefer treatment in primary care rather than specialist services that reside within secondary care (e.g. Alcohol Care Teams in hospital gastroenterology units) or community settings (e.g. community alcohol services) which they perceive as stigmatising [19], and primary care is at least as effective as specialist services for patients with low to moderate dependence [20]. However, pharmacotherapy prescriptions are usually initiated in a specialist service and general practitioners may feel that they lack the expertise in utilising pharmacotherapy, meaning that fewer patients are prescribed medications which could assist in the management of alcohol use disorders in primary care.

Liverpool is a city with the 3rd highest prevalence (2.53 cases/100,000) of alcohol dependence in England [21] and is also ranked as the 3rd most deprived local authority out of 317 across England [22]. Our previous work has identified decreasing levels of identification of alcohol dependence in primary care over the last five years, with areas of low socioeconomic status having higher prevalence, but lower levels of pharmacotherapy prescribing for those patients once diagnosed [23]. Understanding the potential facilitators and barriers to accessing management and support for alcohol dependence within primary care is key to informing how practitioners can be supported to fulfil their role and meet patient support needs. The present study aims to identify perceived barriers and facilitators to treatment access and entry in primary care to allow us to identify future needs for such services.

## Method

### Participants and recruitment

Patients were eligible to take part if they:

i.  Were aged over 18 years;

ii.  Had a current diagnosis of alcohol dependence;

iii.  Were registered with a GP surgery in Liverpool, UK.

Patients with any other diagnosed co-occurring substance use disorder were excluded from the study as treatment needs of this group differ from those who only present with alcohol dependence [9].

Health Care Professionals (HCPs) were eligible to take part if they:

i. Had previous or current experience of management, treatment, or referral of adults with alcohol dependence in Liverpool.

Twenty-Six participants were recruited from three NHS sites in Liverpool (one primary care general medical practice, registered patients aged over 18: 50,037; one secondary care inpatient setting for patients with substance use disorders; one secondary care specialist Alcohol Care Team in a general hospital), where authors NvG and RY (Associate GP; Homeless Lead), CK (Consultant Psychiatrist) and LO (Nurse Consultant) were respectively based. The multi-site approach and broad participant inclusion criteria were intended to allow for participation of patients and staff from a variety of treatment settings, including but not limited to primary care, as a significant part of treatment for patients with alcohol dependence currently takes place outside of primary care, such as in hospitals or specialist alcohol treatment services.

A multimethod purposive sampling strategy was used, which involved advertising the study to participants who met the eligibility criteria. Patients (N = 11) were recruited via the Liverpool GP practice and inpatient setting. GP registered patients with a SNOMED code for alcohol dependence [10] who met the remaining inclusion criteria were identified by practice staff and sent an SMS (N = 357) with brief information about the study and a link to the information sheet with contact details for the lead researcher (CS). CS also attended the inpatient setting in person, where staff identified eligible patients to invite to participate.

To recruit HCP participants (N = 15), authors NvG, CK and LO circulated information about the study, including the Participant Information Sheet and the researcher's (CS) contact details via email to their colleagues, and spread this information via word-of-mouth. In addition, CS also completed face-to-face recruitment at the inpatient setting and the general hospital. See Table 1 for participant demographic characteristics.

**Table 1. Demographic details of participants (patients and staff).**

|  | Patients (n = 10) | Staff (n = 15) |
|---|---|---|
| **Gender, n (%)** |  |  |
| Male | 5 (50.0) | 7 (46.8) |
| Female | 5 (50.0) | 8 (53.3) |
| **Age, mean** | 40.00 | 40.71 |
| **Ethnicity, n (%)** |  |  |
| White British | 10 (100.0) | 12 (80.0) |
| White Irish | 0 | 1 (6.7) |
| British Asian | 0 | 1 (6.7) |
| Mixed (Asian White) | 0 | 1 (6.7) |
| **Place of residence, n (%)** |  |  |
| Liverpool | 10 (100.0) | 5 (33.3) |
| Other areas | 0 | 10 (66.7) |
| **Recruited from:** |  |  |
| GP surgery | 10 | 4 |
| Inpatient treatment setting | 1 | 4 |
| Alcohol Care Team | 0 | 7 |
| **Occupation, n (%)** |  |  |
| Unemployed | 7 (70.0) | 0 |
| Employed | 2 (20.0) | 25 (100) |
| Student | 1 (10.0) | 0 |
| Nurse | 0 | 9 (60.0) |
| Nurse assistant | 0 | 2 (13.3) |
| GP | 0 | 3 (20.0) |
| Psychiatrist | 0 | 1 (6.7) |

## Ethical approval

This study was approved by the Health Research Authority; IRAS project ID: 313497; REC reference: 22/LO/0335. Sponsor: Liverpool John Moores University.

## Data collection

Semi-structured one-to-one interviews were conducted by an experienced researcher with all participants, either face-to-face at Liverpool John Moores University (LJMU) (N = 6), on NHS premises (N = 5) or in a public place (N = 3), by phone or online via Microsoft Teams (N = 11), depending on participant preferences. Please see S1 File for the patient interview guide and S2 File for the Health Care Professional interview guide. Participant invites were sent out between May 30th and June 21st 2022, and interviews took place between May and July 2022. Following the provision of written informed consent, interviews began with the collection of demographic data (gender, ethnicity, age, place of residence and occupation) and then addressed participants' experiences of alcohol treatment or treatment provision, particularly challenges and facilitators to treatment access with some differences between questions posed to staff and patients. All interviews followed an interview guide, which was developed by drawing on current literature, consultation with study collaborators including healthcare professionals, academic collaborators, and expert-by-experience collaborator (MR).

Twenty-three interviews were audio-recorded with participants' consent. Since two participants preferred not to be recorded, written notes were taken of these interviews. Conversations lasted between 23 and 81 minutes and participants received a high street (Love2Shop) voucher with a value of 25 GBP (National Institute for Health Research involve rate for participant and patient involvement) as a thank you for their participation in the study. Audio recordings were assigned participant numbers and transcribed verbatim onto Microsoft Word documents removing any identifiable information.

## Patient and public involvement

NHS clinicians such as GPs, nurse consultants and psychiatrists and a patient and public involvement collaborator were involved in the development of the project. The group has met regularly throughout the project. Before the interviews began, they met to discuss the project priorities and the areas of safety netting and alcohol care that were of interest to patients. This was considered when writing the interview schedules. Since the completion of the interviews, the group has met to discuss the findings and highlight important results from the multiple stakeholders perspective.

## Data analysis

Three researchers (CS, MM and CH) coded transcripts manually and subjected them to thematic analysis according to Braun and Clarke's six steps [24] while data collection took place, as well as afterwards. The analytic approach was inductive and data driven. Each researcher coded different transcripts separately but to ensure triangulation, three interviews were coded by all three researchers. Following this, the authors compared and discussed codes with each other, then identified potential recurring themes from lived experiences and expectations of current primary care treatment pathways for alcohol dependence separately before coming back together to compare and revise these, first amongst themselves and then again together with co-investigator PS, who has extensive experience in thematic analysis to establish procedural reliability and conceptual credibility [25]. The iterative coding process enabled the continual revision of themes until the final classifications of major themes were agreed by the

team. During meetings, frequent comparisons were made across codes and the interview data to develop, review, and refine themes based on the complementarity, convergence, and dissonance of ideas across data sources [26,27]. While we had intended to interview 15 individuals from the patient group, our analyses indicated that data saturation was reached with 11 patient participants, as this sample was fairly homogenous. All findings were reviewed within the research group and any disagreements were resolved by discussion. The data was then reanalysed in context of these themes. Summary qualitative interviews and resulting themes can be accessed on the LJMU open access data repository for this study via this link https://doi.org/10.24377/LJMU.d.00000157.

## Results

Interviews were conducted with 10 patients with alcohol dependence (one participant was excluded after indicating they would like to take part as it became clear that they lacked capacity to consent) and 15 staff members (see Table 1 for participant characteristics).

As shown in Fig 1, we identified concerns (subthemes) among staff and patients that related to these three parts of the treatment journey.

### Point of access

**Reactive, not preventative.** By the time individuals were diagnosed with alcohol dependence, some people had developed significant physical ailments resulting from their alcohol consumption. These included "stomach problems", jaundice and ascites, and prompted them to seek help from primary care or the local Community Alcohol Service. The remaining individuals had only received a diagnosis following physical or mental emergencies or involvement

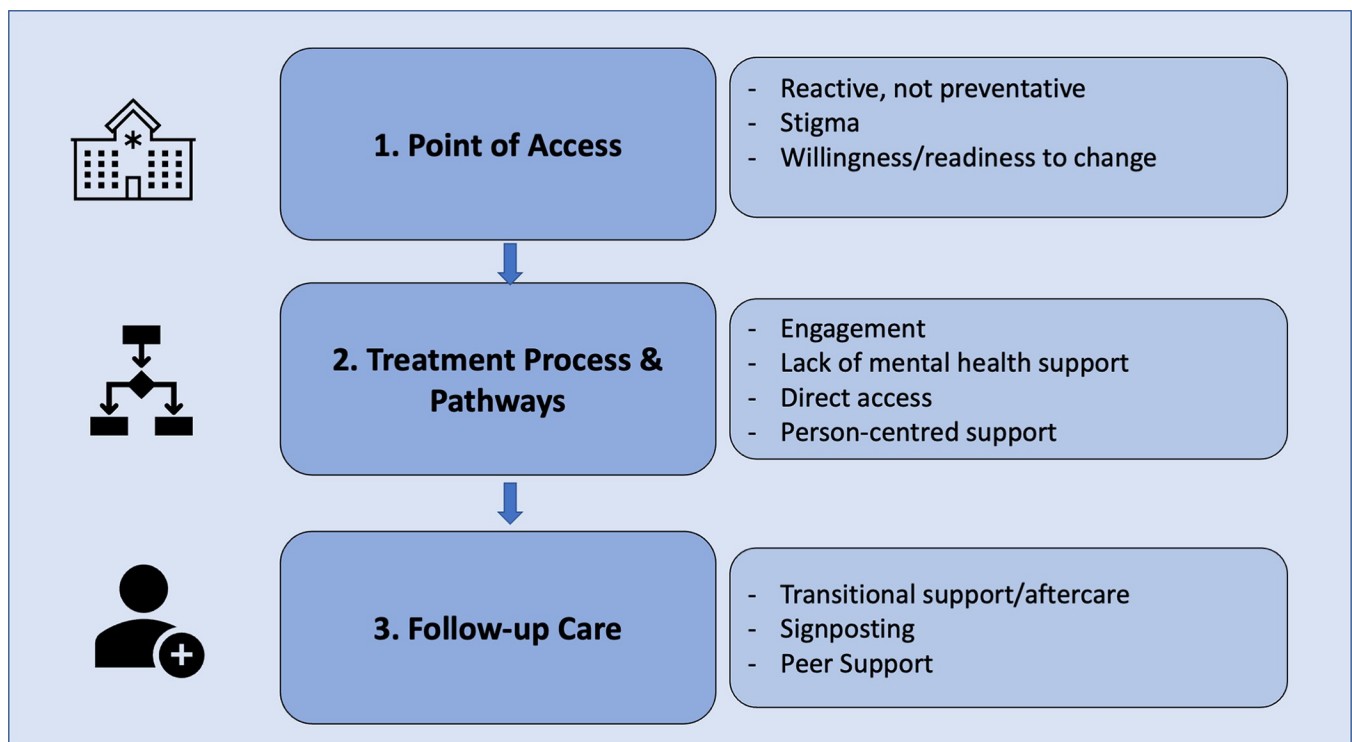

**Fig 1. Overview of themes.**

with the criminal justice system for alcohol-related reasons. This was a sentiment echoed by all of the patient participants and as one participant illustrates, primary care adopted a reactive rather than preventative approach when it came to diagnosing and treating him:

"*What happened was I went to the GP because I couldn't keep anything down. [. . .] I had ascites of the liver. I was getting treated for constipation for weeks. So, it [alcohol dependence] never got picked up on for about eight weeks and then one day I just went the surgery and I was bright yellow, my legs were the size of tree trunks and my stomach was [swollen]. [. . .] And I just said I am not leaving this surgery until you do something, and I received a letter to go over to the hospital to be admitted.*" (Participant 11, patient)

**Stigma.** All patients and some staff clearly identified the stigma surrounding alcohol dependence as a barrier to accessing and using treatment. More specifically, patients highlighted, for example, difficulties identifying their problems as alcohol dependence, which arose from a contradiction between the stereotypical image of people with alcohol dependence and their self-perception:

"*I was 19 just sat in a room with people two to three times my age, all big red noses. Basically, like what you classify as, like, your standard alcoholic and I was sat there as this 19-year-old student. I had alcohol in my bag at the time, but in my mind, I'm not the same of these people.*" (Participant 18, patient)

Furthermore, some clinicians and all patients made reference to a fear or an expectation of judgment by healthcare staff that can prevent access to treatment, though individual health care staff did not perceive that they were the staff holding stigmatising attitudes:

"*So, like, the stigma of alcoholism is something that's kind of always made me shy away from a lot of treatment. So just simply, the act of talking about it, and not being judged, is a huge thing.*" (Participant 16, patient)

"*They [people with alcohol dependence] do feel very judged, but they feel judged by society in general, you know, just, you know, if, if the clothes are dirty, you know, they generally worry if they smell, they don't like sitting in crowded rooms and things like that.*" (Participant 9, general nurse)

**Willingness/Readiness to change.** Over half of the clinician interviews showed an expectation for people to possess the will or readiness to accept and commit to treatment, without which efforts to engage them would likely be unsuccessful:

"*To be honest, it's on them then isn't it, because you are agreeing dates and times and they're choosing not to pick up the phone or- If you wanted it that much, then you'd apply yourself to it wouldn't you?*" (Participant 20, mental health nurse)

While some staff and most patients agreed that help seeking, for example, by contacting or attending an alcohol service is a sign of the patient's motivation to recover, people's experiences suggest that there does not appear to be a clear consensus on what constitutes willingness or readiness to change:

"*I think, yeah, genuinely, most of our patients do want the help, otherwise they wouldn't come to clinic appointments, and they wouldn't attend an A&E unless they were severely unwell.*" (Participant 27, nurse)

*"They [alcohol service] were like, you need to show willing to get sober and whatever before we want to consider putting you into inpatient detox. We don't really want to put you to an inpatient detox. [. . .] The fact that I'm phoning up is showing willing in my opinion."* (Participant 12, patient)

## Treatment process and pathways

**Engagement.** Both participant groups acknowledged that engaging with alcohol services or primary care for the first time posed difficulties. Outreach, that is a proactive approach involving healthcare staff seeking out or contacting people in the community as well as relationship-building between staff and individuals with alcohol dependence were identified as facilitating engagement:

*"If someone just occasionally said "come in for a welfare check" [. . .],er to talk about your drinking, your mental health. Erm, that would be the ideal, rather than me not getting around to doing it for a long time."* (Participant 2, patient)

*"And it's that ability, I think, to then go in as the same person, as the person who knows their history, knows where [the person is] at, understand him [. . .] We were able to head off most of the damage by being reactive and knowing the person that we're dealing with as well."* (Participant 1, GP)

**Lack of mental health support.** Several participants draw attention to the neglect of mental health treatment in favour of a focus on medical and emergency treatment of alcohol dependence. The requirement established by mental health services for people to achieve abstinence or sobriety before being able to access mental health support is highlighted as a barrier to treatment:

*"And then if people are drinking then, they'll [psychological services] just be like "well, they've got a drinking issue, we can't speak to them until they resolve that."* (Participant 23, nurse)

**Direct access.** Among both participant groups, delayed access to alcohol services as well as the need to complete several steps to receive the desired treatment is criticised whereas straightforward and speedy access, for example through a drop-in or self-referral is perceived as facilitating treatment entry. This was a particularly prevalent view in the patient participant group:

*"Going through multiple, multiple, steps to do something, you lose the will to engage"* (Participant 13, patient)

*"It's all gotta be like "oh well, we'll send this information back to your GP and we'll refer you to these and then-"[. . .] That could take weeks. Instead of you thinking "right, I'm in crisis. [. . .] I know that there's this place I can go to. I can just go."* (Participant 19, patient)

**Person-centred support.** Experiences of being directed to or offered treatment that did not suit their individual needs are common among patients, with all patients talking about the need for bespoke care according to their needs. On the contrary, a holistic and personal healthcare service that considers their physical and mental health, and helps them pick and access the right treatment is valued and desired:

"*Every single experience I've ever had they've said, 'do X, Y, and Z before we'll even be willing to do that, so first of all, let's try and reduce [your alcohol intake] or we'll give you this medication'. But when you look at my medical records, [. . .] none of that's ever worked. I need to be in an isolated community for the foreseeable future.*" (Participant 18, patient)

"*So the job and remit for the alcohol worker was to help the people with alcohol problems, navigate, the multiple alcohol services that are available, so ensuring that the person with alcohol problems got to the right place, which worked absolutely brilliantly*" (Participant 6, GP)

## Follow-up care

**Transitional support/Aftercare.** Most interviewees (all patients and over half of the staff) identified a lack of aftercare in the community and rehabilitation following detoxification, which they feel is linked to an increased likelihood of relapse. Challenges for people in accessing follow-up care included waiting times for rehabilitation and inability to finance this privately to decrease wait time. Finally, the data revealed a reluctance among GPs to prescribe anti-craving medication:

"*We can detox the patient, we can put them on anti-craving medication. But afterwards, what is there for patients? And to be fair I do think [name of city] is quite good. We do have a lot of services.*" (Participant 29, nurse)

"*I certainly think it's a good thing that we can now prescribe it [anti-craving medication] but I personally wouldn't totally feel comfortable initiating it without them having seen some alcohol services first and got the OK that they think it's the right thing to prescribe. Unless they've been on it before.*" (Participant 08, GP)

**Signposting.** Individuals with alcohol dependence expressed a need to be informed and signposted to alcohol and/or mental health services, with some interviewees pointing to an avoidable deterioration in their dependence as a result of delayed signposting following an alcohol-related hospital admission. Both primary and secondary healthcare professionals interviewed described signposting and/or referrals as part of their routine practice. However, four people with alcohol dependence and three clinicians appeared to agree that awareness of alcohol services among healthcare professionals, particularly GPs could be improved:

"*It probably isn't hidden but when you're drinking-, I didn't even know about AA meetings and stuff. [. . .] it's not talked about or services that are available and- like there's no- there's not much information anywhere.*" (Participant 19, patient)

"*I think you can see a lot of leaflets in the GP surgery, but I would place a bet that the GP doesn't know what half of them leaflets are for.*" (Participant 11, patient)

"*The majority of people who come here [alcohol charity] are not referred by GPs or medical practitioners. They come via social services, probation, or residential detox centres.*" (Participant 11, volunteer for alcohol charity)

**Peer support.** Support provided by other individuals in recovery -whether in the form of one-to-one counselling or group support- is associated with feeling understood and connected through shared experience, which helped interviewees to maintain abstinence or work towards it after relapse:

"*Talking to people that have been there and done it and wore the t-shirt [AA t-shirt], that's a very comforting thing because if you hear stories you identify with people.*" (Participant 5, patient)

## Discussion

### Summary

Primary care services have a key role in supporting adults with alcohol dependence and require appropriate provision of services. In this study we examined the perceptions of primary care practitioners and adults with alcohol dependence, regarding service provision and help seeking behaviours for adults with alcohol dependence. Three inter-related themes reflected the perceived context-specific facilitators and barriers to treatment access for alcohol dependence from the perspective of patients and staff. The first theme, *Point of Access* related to current service provision being reactive rather than preventative, the stigma associated with an alcohol dependence and a person's willingness or readiness to change. The second theme identified the *treatment process and pathways* and highlighted the different levels of engagement, lack of mental health support, and the importance of direct access and person-centred support. The third theme discusses *follow-up care* and the transitional support or aftercare for alcohol dependence including signposting and peer support.

### Strengths and limitations

The strength of this study is that it was possible to gather experiences of people with alcohol dependence and staff working in this area in a city with high levels of deprivation, and recurrence of themes indicated that data saturation was reached using the small number of patients and health care professionals recruited in the study. This study has been able to report on the barriers and facilitators for patients and staff when treating alcohol dependence in primary care. Another strength is that interviews and coding of data were carried out by three researchers, thus reducing any preconceptions of one researcher that may influence how data is coded and interpreted [27]. To mitigate this, synthesis and interpretation of themes were discussed extensively and agreed by the research team and with the patient and public involvement collaborator.

The study is limited as the results were based in this one area of England and may not reflects barriers and facilitators to accessing treatment across the UK and internationally. All of the patient participants identified as white ethnicity, and while this reflects the majority of alcohol dependence diagnoses in Liverpool [23], it cannot tell us about the barriers faced by ethnically diverse groups. Future research recruiting from a wider population than the primary care settings that we utilised for patient recruitment could allow a more diverse representation of people with alcohol dependence. In addition, we focussed our interviews on people with alcohol dependence and health care professionals with experience in their management; it would be interesting to also gain insight from hazardous and harmful drinkers and health care professionals who have yet to be involved in the active management of people with alcohol dependence to understand the potential barriers and facilitators from these groups. We did not systematically collect data on history of alcohol use and severity of alcohol dependence during this study, and thus we cannot identify barriers and facilitators that may be relevant for shorter-term and longer-term dependence.

### Comparison with existing literature

General Practitioners voiced concerns about the lack of training in identifying alcohol-related problems, with estimates that the training in medical curricula had amounted to a single

lecture. This has also been demonstrated in previous studies in Scotland [18], where GPs stated that alcohol training was poor and delivered too early in the curriculum. It is noteworthy that low identification and limited support of people with alcohol use disorders is a historical problem. In the 1970s in the UK, attempts were made to investigate the apparent unwillingness of primary care practitioners to identify and engage with individuals with alcohol use disorders [28]. The body of work resulting from the Maudsley Alcohol Pilot Project (MAPP) identified three factors that influence primary care practitioners' therapeutic attitudes towards people with alcohol use disorders and thus their commitment to treat (termed "therapeutic commitment") this patient group [29] The first factor, Role Adequacy–the extent to which primary care practitioners feel that they are adequately prepared for the role—has also been identified in the current study whereby GPs felt that they lacked specific training and knowledge for identification and diagnosis of alcohol use disorders. The second factor of role legitimacy–how much primary care practitioners regard the identification and management of alcohol use disorders as being their responsibility–can result in an apathy towards the treatment of alcohol use disorders in primary care. The third factor, role support–how much primary care practitioners feel supported by their co-workers to identify and diagnose alcohol use disorders–was identified by some patients and practitioners in the present study. For example, signposting between services was identified by both patients and GPs, with many stating that they/their GP were unaware of the various organisations that could provide care for patients with complex needs. Good signposting to relevant groups has been noted as a facilitator to treatment access in previous research [18,30]. Cartwright and co-workers [28,29,31] suggest that to successfully develop community-based approaches to the management of alcohol use disorders, primary care practitioners must have access to experience and support to help them develop more positive therapeutic attitudes. However, such attitudes seem more resistant to change within services than the information giving observed in e.g. signposting practices, with patients reporting stigma as a significant barrier to accessing treatment. In the present study, there was evidence that patients experienced internally driven stigma whereby they struggled accepting the identity of being "*an alcoholic*", which has been demonstrated in previous research in non-treatment seeking hazardous drinkers [20] and substance users [32]. Moreover, high perceived alcohol-related stigma reduced likelihood of treatment seeking [33]. There was also evidence that some HCPs used potentially stigmatising language which could further distance treatment seekers from treatment entry [18].

## Implications for research and/or practice

Importance of alcohol training has been noted in this study and in previous research [34], with the need for ongoing support and experienced mentors in community settings to increase the positive therapeutic commitment of primary care practitioners [31]. By identifying treatment barriers and facilitators through interviews with patients and healthcare professionals and triangulating the themes with our lived experience group, we have developed recommendations for better treatment provision and pathways (see Fig 2). The results of this study have helped us to revise and plan the pilot of the recommendations for treatment pathways in areas of Liverpool with higher levels of alcohol dependence and social deprivation. The NICE [9] clinical guideline for managing alcohol dependence identifies using a person-centred approach as a key principle. There is evidence in Scotland that using a Primary Care Alcohol Nurse Outreach Service (PCANOS) is effective at removing many of the barriers identified by the current study (lack of patient engagement, lack of GP resource, skills, time, stigma) [35] and adoption of this model in primary care in Liverpool could significantly improve identification and engagement of these patients in PC. Other innovative models developed for primary care settings (e.g. The 15-method) [36] could also provide structure for a stepped-care model to treat alcohol use

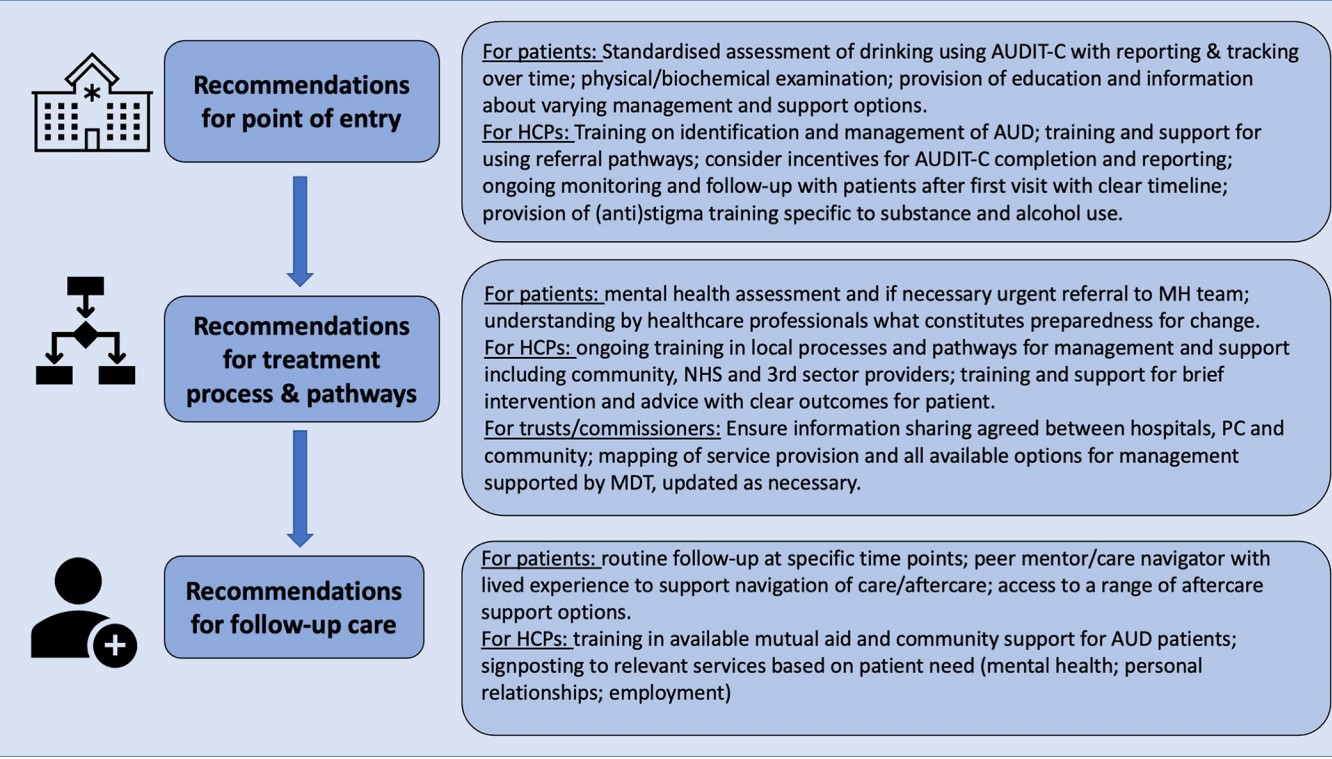

**Fig 2. Recommendations for practice.**

disorders in the Liverpool City Region, and has demonstrated acceptability and feasibility with both GPs and patients. This research is important as it will improve care for individuals with alcohol dependence and could reduce the number of alcohol related hospital admissions to reduce costs to the NHS. Implementing the suggested changes in deprived areas of Liverpool (a follow-up project) will help to reduce health inequalities by decreasing the gap between needed treatment and treatment access, among people with alcohol dependence. This will have positive effects on the care of people with alcohol dependence, their quality of life and health through a reduction in alcohol-related harm.

## Supporting information

**S1 Checklist. STROBE statement—Checklist of items that should be included in reports of observational studies.**
(DOCX)

**S1 File. HRA approved interview topic guide for service users.**
(PDF)

**S2 File. HRA approved topic guide for health care professionals.**
(PDF)

## Acknowledgments

The authors would like to thank all staff members and people with alcohol disorders who took the time to be interviewed during a very busy period for primary care. The authors would also

like to thank the patient and public involvement group for their guidance and contribution to this and the wider study.

## Author Contributions

**Conceptualization:** Catharine Montgomery, Pooja Saini, Melissa Rice.

**Data curation:** Christine Schoetensack, Molly McCarthy, Claire Hanlon.

**Formal analysis:** Catharine Montgomery, Pooja Saini, Christine Schoetensack, Molly McCarthy, Claire Hanlon.

**Funding acquisition:** Catharine Montgomery, Lynn Owens, Cecil Kullu, Nadja van Ginneken, Melissa Rice.

**Investigation:** Catharine Montgomery, Christine Schoetensack.

**Methodology:** Catharine Montgomery, Pooja Saini, Lynn Owens, Nadja van Ginneken, Melissa Rice, Ryan Young.

**Project administration:** Catharine Montgomery, Christine Schoetensack, Cecil Kullu, Nadja van Ginneken, Ryan Young.

**Supervision:** Catharine Montgomery, Pooja Saini, Lynn Owens, Cecil Kullu, Nadja van Ginneken, Ryan Young.

**Writing – original draft:** Catharine Montgomery, Christine Schoetensack.

**Writing – review & editing:** Catharine Montgomery, Pooja Saini, Molly McCarthy, Claire Hanlon, Lynn Owens, Cecil Kullu, Melissa Rice, Ryan Young.

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
