## [Decision Letter · Decision Letter 0]

20 Jun 2023

PONE-D-23-12922Improving access to treatment for alcohol dependence in primary care: A qualitative investigation of factors that facilitate and impede treatment access and completion.PLOS ONE

Dear Dr. Montgomery,

Thank you for submitting your manuscript to PLOS ONE. After careful consideration, we feel that it has merit but does not fully meet PLOS ONE’s publication criteria as it currently stands. Therefore, we invite you to submit a revised version of the manuscript that addresses the points raised during the review process.

 We recommend that authors use the COREQ checklist, or other relevant checklists listed by the Equator Network, such as the SRQR, to ensure complete reporting (http://journals.plos.org/plosone/s/submission-guidelines#loc-qualitative-research).  Please submit COREQ checklist with your resubmission. 

We look forward to receiving your revised manuscript.

Kind regards,

Julia Morgan

Academic Editor

PLOS ONE

“This work was supported by an NHS Liverpool Clinical Commissioning Group Research Capability Fund grant RCF21-22/07 to CM, PS, LO and CK.

The funders played no role in the design, data collection, analysis, decision to publish or preparation of the manuscript.”

Reviewers' comments:

Reviewer's Responses to Questions

**Comments to the Author**

1. Is the manuscript technically sound, and do the data support the conclusions?

Reviewer #1: Partly

Reviewer #2: Yes

2. Has the statistical analysis been performed appropriately and rigorously? 

Reviewer #1: N/A

Reviewer #2: Yes

3. Have the authors made all data underlying the findings in their manuscript fully available?

Reviewer #1: Yes

Reviewer #2: Yes

4. Is the manuscript presented in an intelligible fashion and written in standard English?

Reviewer #1: Yes

Reviewer #2: Yes

5. Review Comments to the Author

Reviewer #1: Issues of alcohol treatment have received little research attention over recent years; so, it is encouraging to read this paper. It tackles a long-standing problem of improving access to alcohol treatment and addressing the barriers. The paper provides a sound rationale for the research, the research procedures and methods are well described, and analytical methods are appropriate. Attention to the following is needed:

• There is lack of clarity in the use of the terms ‘primary care’ and ‘GP’. For readers who are not UK based (and even for those who are) – please define ‘primary care’ indicating the role of the GP in the primary care system. Specialist alcohol treatment services are mentioned – where do they sit in the ‘primary care’ system?

• The sample: recruited via a GP service and inpatient services – how many from each of the 3 types of services? It is explained that the sample is purposive but the possible effects of stakeholder selection on the information collected is not considered in the limitations. Specialist alcohol treatment services are mentioned (L125) but why were community-based services not accessed? It is a very small sample so limitations, even in the Liverpool area, need to be discussed. Why did the sampling stop at 11 patients – possibly data saturation? How was this determined? Would non-white groups have been found in third sector services? Some explanation for the number is needed.

• In reporting the findings, we are provided with good quotes to illustrate the themes. However, we do not get an overall picture of what the participants reported. I do not want to ask for a quantitative approach in what is a qualitative study, but it would be useful to know how many patients and/or staff thought X or Y, and what level of agreement/ disagreement was found where patients and staff had different views.

• The discussion needs to be extended – the section on comparison with existing literature in particular. The journal does not impose a word limit. In particular, the discussion lacks contextualisation of the work against a long history of attempts to improve the management of alcohol problems and access to treatment in primary care (especially with a focus on GPs). There is literature on the lack of focus in undergraduate medical curricula as well as in post-graduate training (apart from the Scottish example provided). Without going into detail, some recognition of seminal work on the lack of ‘therapeutic commitment’ (Cartwright and Shaw) and subsequent attempts (e.g. community alcohol teams, CATs) could be made and some discussion of how some barriers appear to resist change while other factors such as ‘signposting’ have entered the picture with the growing complexity of what counts as ‘primary care’ services. While the paper makes a good contribution for developing services in Liverpool, an extended and contextualised discussion would help to broaden the relevance of the work for wider consideration.

Reviewer #2: Montgomery et al. conducted qualitative research with individual interviews in 10 adults with alcohol dependence and 15 primary care treatment providers to identify perceived barriers and facilitators to treatment access and entry in primary care. The authors addressed a highly relevant topic and provided new results. The research methods are generally valid, and the manuscript is written very clearly and concisely.

Major comments

1. Please provide a clear definition of “alcohol dependence” as used in this article. Ideally, also differentiate other forms of problematic drinking when used, such as “hazardous alcohol use”.

2. In the introduction, please include information on guideline recommendations and primary care as usual for AD in the UK. In this context, please elaborate which role GPs and other primary health care providers exactly are supposed to have in relation to diagnosing/treating AD and other forms of problematic drinking such as harmful and hazardous drinking. For example, I imagine that GPs should identify harmful/hazardous drinking early and provide brief advice, but should refer people with AD to specialised treatment? Who is supposed to prescribe pharmacotherapy? What behavioural support is offered to whom?

3. Please justify the following methodological choices:

a. Patients “had a current diagnosis of AD”. Why did you not attempt to include hazardous/harmful drinkers? In the light of prevention, it would be very interesting to know why these people do not use primary care services.

b. “Patients with any other diagnosed co-occurring substance use disorder were excluded…” How high was this proportion? What implications does this exclusion have for the generalisability of your findings?

c. HCPs were only included if they “had previous or current experience of management, treatment, or referral of adults.” Why were those with no experience excluded? This is also an important group - why are they not active in helping patients drink less?

4. The Limitations paragraph should be extended in light of comment #3.

Minor comments

5. Introduction: “Screening and interventions to address hazardous alcohol use are utilised in primary care, …” – please provide a few concrete examples.

6. Methods: please provide some more information on the following.

a. “Patients … were identified by practice staff and sent an SMS” – really? Is this a common approach in the UK? I’ve never heard that patients were contacted for research this way. What was you experience with this approach?

b. “Semi-structured one-to-one interviews were conducted … by phone or online via Microsoft Teams (N = 11), depending …” How did the data collection differ between face-to-face interviews and interviews via phone and Teams?

7. Methods: please provide the full interview guide as supplementary material.

8. Results. With regard to the details of patients (Table 1), it would have been interesting to know at what stage of AD they were, since how long that had been diagnosed, what treatments they had used.

9. Results, line 210: “for about for about” duplicate.

10. Table 1: use one decimal throughout.

11. Discussion: “inability to finance the latter” – briefly explain what services patients need to pay for in the UK.

12. I like Figure 1! In Figure 2, however, it is not clear whether the boxes in rows 2 and 3 also contain recommendations. The points mentioned here are largely the same as in Figure 1.

13. Use consistent language: service user or patient?

14. Carefully check all acronyms throughout the paper. All acronyms should be explain when used for the first time. Do only use acronyms for words used at least 5 times. I recommend not to use an acronym for AD.

6. PLOS authors have the option to publish the peer review history of their article (what does this mean?). If published, this will include your full peer review and any attached files.

Reviewer #1: No

Reviewer #2: **Yes: **Daniel Kotz

---

## [Author Response · Author response to Decision Letter 0]

20 Jul 2023

PLEASE SEE SUBMITTED RESPONSE TO REVIEWERS FILE AS FORMATTING HAS BEEN LOST IN THIS TEXT BOX

Reviewer #1: Response Location in manuscript

Issues of alcohol treatment have received little research attention over recent years; so, it is encouraging to read this paper. It tackles a long-standing problem of improving access to alcohol treatment and addressing the barriers. The paper provides a sound rationale for the research, the research procedures and methods are well described, and analytical methods are appropriate. We thank Reviewer 1 for this positive summary comment. N/A

Attention to the following is needed:

There is lack of clarity in the use of the terms ‘primary care’ and ‘GP’. For readers who are not UK based (and even for those who are) – please define ‘primary care’ indicating the role of the GP in the primary care system. Specialist alcohol treatment services are mentioned – where do they sit in the ‘primary care’ system?

 We have defined what the primary care system is and how GPs are within the primary care system as a first point of contact for NHS services in the introduction: 

“The primary care system in the NHS provides a first point of contact with the health care system and includes General Practitioners (GPs) of medicine, pharmacists, dentists and opticians. Over 90% of the UK population is registered with a General Practitioner [10], providing General Practitioners with the opportunity to assess and identify patients with alcohol use disorders [11].”

Specialist alcohol services (usually as part of the secondary care gastroenterology team) are not within the primary care system but receive referrals from primary care to treat people with alcohol dependence once they have been identified by a GP in the primary care system. We have tried to make this clear in the introduction and throughout the manuscript, e.g. 

“Existing evidence suggests that patients prefer treatment in primary care rather than specialist services that reside within secondary care (e.g. Alcohol Care Teams in hospital gastroenterology units) or community settings (e.g. community alcohol services) which they perceive as stigmatising [16], and primary care is at least as effective as specialist services for patients with low to moderate dependence [17].”

“Twenty-Six participants were recruited from three NHS sites in Liverpool (one primary care general medical practice, registered patients aged over 18: 50,037; one secondary care inpatient setting for patients with substance use disorders; one secondary care specialist Alcohol Care Team in a general hospital)…” Lines 85-87; 105-112; 138-141. 

The sample: recruited via a GP service and inpatient services – how many from each of the 3 types of services? It is explained that the sample is purposive but the possible effects of stakeholder selection on the information collected is not considered in the limitations. Specialist alcohol treatment services are mentioned (L125) but why were community-based services not accessed? It is a very small sample so limitations, even in the Liverpool area, need to be discussed. Why did the sampling stop at 11 patients – possibly data saturation? How was this determined? Would non-white groups have been found in third sector services? Some explanation for the number is needed.

 We apologise for neglecting to include this information initially, and thank Reviewer 1 for bringing this to our attention. We have added the types of service that HCP and patient populations were recruited from in Table 1. It is worthy of note that the majority of patients were contacted via the GP surgery as we were interested in the barriers to primary care patients in our study. No patients were recruited from the Alcohol Care Team as this team provides ambulatory detoxification before referral to the community alcohol service and in our previous studies, patients in this service are usually on a tight time scale and want to leave the hospital setting as soon as possible. 

The patient sample was fairly homogenous, and recruitment ceased at 11 patients due to data saturation and this has been clarified in the method section. 

Community based services were not accessed as part of this study because we were interested in low levels of identification in primary care. However, we have noted this as a limitation in the limitations section, in addition to the possibility of a more diverse sample being available across other recruitment settings. Table 1 (line 180+). 

Lines 225-227; 407; 418-424. 

In reporting the findings, we are provided with good quotes to illustrate the themes. However, we do not get an overall picture of what the participants reported. I do not want to ask for a quantitative approach in what is a qualitative study, but it would be useful to know how many patients and/or staff thought X or Y, and what level of agreement/ disagreement was found where patients and staff had different views.

 We agree with Reviewer 1 that this is a useful descriptive analysis, and we have tried to include statements to reflect the degree of agreement within the groups and between the two groups. There was a great deal of homogeneity within the patient cohort with agreement on barriers and facilitators being high. There was a slightly lower level of agreement between the HCP, which likely reflected their roles in the services where they worked. There was some agreement between patients and staff, but also many points where they did not agree. Highlighted phrases throughout results section. 

The discussion needs to be extended – the section on comparison with existing literature in particular. The journal does not impose a word limit. In particular, the discussion lacks contextualisation of the work against a long history of attempts to improve the management of alcohol problems and access to treatment in primary care (especially with a focus on GPs). There is literature on the lack of focus in undergraduate medical curricula as well as in post-graduate training (apart from the Scottish example provided). Without going into detail, some recognition of seminal work on the lack of ‘therapeutic commitment’ (Cartwright and Shaw) and subsequent attempts (e.g. community alcohol teams, CATs) could be made and some discussion of how some barriers appear to resist change while other factors such as ‘signposting’ have entered the picture with the growing complexity of what counts as ‘primary care’ services. While the paper makes a good contribution for developing services in Liverpool, an extended and contextualised discussion would help to broaden the relevance of the work for wider consideration.

 We thank Reviewer 1 for this comment, and in our desire to write concisely we had missed these important contextual points. We have amended the discussion to contextualise our work against historic and current attempts to improve management of alcohol use disorders in primary care settings, and we think this greatly improved the discussion. We hope that this is now suitable for Reviewer 1. Lines 430-445; 448-453; 464-466. 

Reviewer #2: 

Montgomery et al. conducted qualitative research with individual interviews in 10 adults with alcohol dependence and 15 primary care treatment providers to identify perceived barriers and facilitators to treatment access and entry in primary care. The authors addressed a highly relevant topic and provided new results. The research methods are generally valid, and the manuscript is written very clearly and concisely.

 Thank you. 

Please provide a clear definition of “alcohol dependence” as used in this article. Ideally, also differentiate other forms of problematic drinking when used, such as “hazardous alcohol use”. We apologise for this omission and have now included a definition of hazardous and harmful drinking, and alcohol dependence in the introduction. 

For patient recruitment, the case definition for alcohol dependence developed by Thompson and co-workers was used to enter codes into the electronic GP recording system (EMIS) to identify patients. This definition can be found in Thompson et al’s supplementary information table in their 2017 paper (citation 10 in the manuscript file). Lines 75-79. Line 150. 

In the introduction, please include information on guideline recommendations and primary care as usual for AD in the UK. In this context, please elaborate which role GPs and other primary health care providers exactly are supposed to have in relation to diagnosing/treating AD and other forms of problematic drinking such as harmful and hazardous drinking. For example, I imagine that GPs should identify harmful/hazardous drinking early and provide brief advice, but should refer people with AD to specialised treatment? Who is supposed to prescribe pharmacotherapy? What behavioural support is offered to whom?

 The National Institute for Health and Care Excellence (NICE) guidelines for diagnosis and management of Alcohol Use Disorders contain key priorities for identification and assessment of alcohol use disorders in all NHS funded settings that a patient may have contact with (primary, secondary , tertiary and community care). For primary care services, it is suggested that all HCP should be able to assess and identify harmful drinking and alcohol dependence using recognised tools, and that after identification, dependent on level of drinking, patients should be offered brief motivational advice followed by: 

-Psychological Interventions (e.g. CBT, behavioural therapy, social network and environment based therapies)

-Behavioural couples therapy (mild dependence)

 -Consideration of pharmacotherapy (acamprosate or naltrexone) to maintain abstinence. 

For those drinking over 15 UK units of alcohol per day, assisted withdrawal is recommended, which would usually be delivered using ambulatory BZD detox after an appointment with the alcohol care team, or an inpatient stay at an NHS or community provider. These options should also be followed by behavioural therapy. 

The problem is that in the UK, as in most areas, the behavioural therapy has very long wait lists so even though the NICE guidelines recommend this for people with hazardous drinking up to severe dependence, many people do not receive this care in a timely manner. We have tried to add something to this effect in the manuscript, but we do not want to be seen to be criticising the primary care system which is already over stretched and underfunded. Moreover, to access the behavioural support via the community alcohol team, a patient who has received a referral from their GP is given a telephone number to call. Upon calling, they are sent a letter (sometimes taking weeks) and upon receiving that letter they have to call the service to make an appointment for the start of behavioural therapy (again some weeks or months later). The same system is in place for many patients on completion of detox when moving to behavioural therapy. Thus many patients have relapsed by this timepoint. Lines 85-87; 93-98. 

Please justify the following methodological choices:

a. Patients “had a current diagnosis of AD”. Why did you not attempt to include hazardous/harmful drinkers? In the light of prevention, it would be very interesting to know why these people do not use primary care services.

b. “Patients with any other diagnosed co-occurring substance use disorder were excluded…” How high was this proportion? What implications does this exclusion have for the generalisability of your findings?

c. HCPs were only included if they “had previous or current experience of management, treatment, or referral of adults.” Why were those with no experience excluded? This is also an important group - why are they not active in helping patients drink less?

 a. We decided to focus on individuals who had a current diagnosis of alcohol dependence as their treatment pathway should be more prescribed such that they would be offered all of the elements identified in the NICE clinical guideline. We also wanted to capture the experiences of those at the most extreme end of harmful drinking. However, we agree with Reviewer 2 that not recruiting the hazardous/harmful drinkers has reduced the richness of our insights and we have noted this as a limitation. 

b. We are unsure to the level of those with co-occurring substance use disorders as we did not ask the administrator in the GP practice how many patients were excluded on the basis of a SNOMED code for substance dependence, and we are unaware how many self-excluded after reading the information that they received via SMS. We do not feel that this limits the generalisability of the results, as those with substance use disorders would be treated according to the NICE clinical guideline 51, which could encompass those with comorbid alcohol and substance use disorders. The purpose of our study was to investigate reasons for very low identification and treatment of alcohol use disorders within the primary care system, whence our focus on this group. 

c. We only included those with experience of the diagnosis and management of alcohol use disorders as we thought that those individuals would be better aware of the potential barriers and facilitators faced by people with alcohol dependence in accessing their service, in the same way we included people with a diagnosis of alcohol dependence in the patient group, rather than those with no diagnosis. We agree with Reviewer 2 that HCPs with no current experience of the diagnosis and management of alcohol use disorders are still an important group for helping patients drink less, but in the current study we felt that these people may not have the necessary insight that comes through interaction with this group of people. We have noted this as a limitation. Line 420-424; 

The Limitations paragraph should be extended in light of comment #3. We have noted the above points a and c as limitations. Line 420-424. 

Minor comments

Introduction: “Screening and interventions to address hazardous alcohol use are utilised in primary care, …” – please provide a few concrete examples.

 We have clarified that in the UK in primary care, the AUDIT-C followed by brief advice based on the score range is usually utilised. Line 101-102. 

Methods: please provide some more information on the following.

a. “Patients … were identified by practice staff and sent an SMS” – really? Is this a common approach in the UK? I’ve never heard that patients were contacted for research this way. What was you experience with this approach?

b. “Semi-structured one-to-one interviews were conducted … by phone or online via Microsoft Teams (N = 11), depending …” How did the data collection differ between face-to-face interviews and interviews via phone and Teams?

 Regarding point a., SMS communication with from GPs and other NHS services is common in the UK, with clinic reminders, requests for information (e.g. photos, descriptions of ailments) being sent via the GP SMS service. However, as far as we are aware this is not as common for recruiting participants for research in individual practices, though it is a method used by the Clinical Research Network (an NIHR funded research infrastructure team). We were fortunate as the practice that we were collaborating with for the project is a large city centre practice with high levels of alcohol dependent patients, and a large homeless access clinic. The patients were identified by the practice administrator by searching for patients with a current SNOMED code for alcohol dependence, and sending an SMS with very brief information and a web link to the participant information sheet and contact details of the researcher. 

Regarding b., all interviews followed the same guide and consenting procedures, so the content of the interviews did not differ according to the mode of data collection. Interviews were organised according to participant preference with many HCP participants opting for online data collection as they were fitting the interview in to a busy work schedule and patients wanting face to face contact with the researcher. - 

Methods: please provide the full interview guide as supplementary material.

 We have included the HRA approved interview guides for both the patients and staff as supplementary files and these are referenced in the manuscript file in the methods section. We have provided these as supplementary file S1 and S2 which are referred to in the manuscript file in lines 186-187. 

Results, line 210: “for about for about” duplicate. Thank you for spotting this, we have removed the additional “for about” Line 253

Table 1: use one decimal throughout.

 We have amended the percentages reported in Table 1 to ensure that all values are to 1dp as requested. Table 1, line 180+

Discussion: “inability to finance the latter” – briefly explain what services patients need to pay for in the UK. We have clarified that when we said “inability to finance the latter” we were referencing the fact that patients could reduce waiting times for rehabilitation and after care by paying to have this provided privately. All of these services are technically free on the NHS, but service provision is variable and more affluent people can pay for private services instead of waiting to be allocated to an NHS provider. Obviously this is not feasible for the majority of our sample, and our comment was intended to reflect their frustrations that they have a long wait for after care, which then increases the likelihood of relapse. Line 352

I like Figure 1! In Figure 2, however, it is not clear whether the boxes in rows 2 and 3 also contain recommendations. The points mentioned here are largely the same as in Figure 1.

 We apologise to Reviewer 2 for the errors in Figure 2. There appears to have been a problem when converting the figure to a PNG file, such that it saved a combination of figures 1 and 2 and we uploaded that. We have now included the correct version of figure 2 in the manuscript documents with the recommendations for Treatment pathways and processes and Aftercare. Figure 2 image file. 

Use consistent language: service user or patient? We have reviewed the manuscript and ensured that the terminology is consistent throughout. We opted for patient throughout as this allows consistency with previous research describing participants as patients, and individuals in our sample were from NHS sites and would be defined by those sites as patients and not service users. Highlighted throughout text. 

Carefully check all acronyms throughout the paper. All acronyms should be explain when used for the first time. Do only use acronyms for words used at least 5 times. I recommend not to use an acronym for AD.

 We have reviewed the manuscript for our use of acronyms as advised by Reviewer 2 and have made any necessary amendments to the acronyms throughout and highlighted these in yellow in the tracked version. Highlighted throughout text.

---

## [Decision Letter · Decision Letter 1]

4 Aug 2023

PONE-D-23-12922R1Improving access to treatment for alcohol dependence in primary care: A qualitative investigation of factors that facilitate and impede treatment access and completion.PLOS ONE

Dear Dr. Montgomery,

Thank you for submitting your manuscript to PLOS ONE. After careful consideration, we feel that it has merit but does not fully meet PLOS ONE’s publication criteria as it currently stands. Therefore, we invite you to submit a revised version of the manuscript that addresses the points raised during the review process. I appreciate that you may not have the level of detail about patient participants' histories that the reviewer asks about, and if so, I will not consider that to be a bar to recommending the paper for publication on resubmission, as long as this is acknowledged in the paper (e.g., as a limitation). 

We look forward to receiving your revised manuscript.

Kind regards,

Simon White

Academic Editor

PLOS ONE

Journal Requirements:

Reviewers' comments:

Reviewer's Responses to Questions

**Comments to the Author**

1. If the authors have adequately addressed your comments raised in a previous round of review and you feel that this manuscript is now acceptable for publication, you may indicate that here to bypass the “Comments to the Author” section, enter your conflict of interest statement in the “Confidential to Editor” section, and submit your "Accept" recommendation.

Reviewer #1: All comments have been addressed

Reviewer #2: (No Response)

2. Is the manuscript technically sound, and do the data support the conclusions?

Reviewer #1: (No Response)

Reviewer #2: Yes

3. Has the statistical analysis been performed appropriately and rigorously? 

Reviewer #1: (No Response)

Reviewer #2: N/A

4. Have the authors made all data underlying the findings in their manuscript fully available?

Reviewer #1: (No Response)

Reviewer #2: Yes

5. Is the manuscript presented in an intelligible fashion and written in standard English?

Reviewer #1: (No Response)

Reviewer #2: Yes

6. Review Comments to the Author

Reviewer #1: (No Response)

Reviewer #2: Montgomery et al. have done a very good job in revising the manuscript according to the reviewers’ comments. I sometimes had difficulties understanding whether the manuscript had been revised and where changes, if any, had been made (sometimes no or wrong line numbers were given). Here is some guidance how to improve response to reviewers, in case you are interested: https://www.degruyter.com/document/doi/10.1515/9783110721621-012/html

The authors seem to have overlooked one comment I made:

8. Results. With regard to the details of patients (Table 1), it would have been interesting to know at what stage of AD they were, since how long that had been diagnosed, what treatments they had used.

7. PLOS authors have the option to publish the peer review history of their article (what does this mean?). If published, this will include your full peer review and any attached files.

Reviewer #1: No

Reviewer #2: **Yes: **Daniel Kotz

---

## [Author Response · Author response to Decision Letter 1]

8 Sep 2023

Regarding Reviewer 2’s comment: 

“Results. With regard to the details of patients (Table 1), it would have been interesting to know at what stage of AD they were, since how long that had been diagnosed, what treatments they had used.”

Unfortunately we do not have this level of detail for all participants as some participants were less forthcoming with this information, and it we did not systematically collect this data as it was not a key aim of the study. As per your suggestion we have noted this as a limitation in lines 417-419 of the manuscript.

---

## [Editor Report · Decision Letter 2]

18 Sep 2023

Improving access to treatment for alcohol dependence in primary care: A qualitative investigation of factors that facilitate and impede treatment access and completion.

PONE-D-23-12922R2

Dear Dr. Montgomery,

We’re pleased to inform you that your manuscript has been judged scientifically suitable for publication and will be formally accepted for publication once it meets all outstanding technical requirements.

Kind regards,

Simon White

Academic Editor

PLOS ONE
---

## [Editor Report · Acceptance letter]

12 Oct 2023

PONE-D-23-12922R2 

Improving access to treatment for alcohol dependence in primary care: A qualitative investigation of factors that facilitate and impede treatment access and completion. 

Dear Dr. Montgomery:

I'm pleased to inform you that your manuscript has been deemed suitable for publication in PLOS ONE. Congratulations! Your manuscript is now with our production department. 

Kind regards, 

on behalf of

Dr. Simon White 

Academic Editor

PLOS ONE